# Monocyte distribution width enhances the detection of infection in patients after primary percutaneous coronary intervention

**Sheng-Feng Lin**[1,2,3,4,5]**, Hui-An Lin**[3,6]**, Peter C. Hou**[7,8]**, Hung-Wei Tsai**[3]**, Sen-Kuang Hou**[3,6]*

1 Department of Public Health, School of Medicine, College of Medicine, Taipei Medical University, Taipei, Taiwan, 2 School of Public Health, College of Public Health, Taipei Medical University, Taipei, Taiwan, 3 Department of Emergency Medicine, Taipei Medical University Hospital, Taipei, Taiwan, 4 Center of Evidence-Based Medicine, Taipei Medical University Hospital, Taipei, Taiwan, 5 Department of Medical Research, Taipei Medical University Hospital, Taipei, Taiwan, 6 Department of Emergency Medicine, School of Medicine, College of Medicine, Taipei Medical University, Taipei, Taiwan, 7 Division of Emergency Critical Care Medicine, Department of Emergency Medicine, Brigham and Women's Hospital, Boston, Massachusetts, United States of America, 8 Harvard Medical School, Boston, Massachusetts, United States of America

* 992001@h.tmu.edu.tw

## Abstract

### Background

Monocyte distribution width (MDW) may serve as an infection marker in acute myocardial infarction (AMI) patients undergoing primary percutaneous coronary intervention (PCI), where infection rates range from 2.4% to 16.6%. We evaluated the association of increased MDW levels with infection occurrence and to assess MDW-based models for predicting infection risk and prolonged hospital length of stay (LOS) ≥ 7 days.

### Methods and results

This retrospective cohort study included AMI patients undergoing PCI at a Taiwanese tertiary teaching hospital from January 1, 2020, to September 30, 2021. Logistic regression models incorporating MDW, Quick Sequential Organ Failure Assessment (qSOFA) score, age, and C-reactive protein (CRP) levels were compared to the Canada Acute Coronary Syndrome (C-ACS) score and stress hyperglycemia ratio (SHR). Among 252 patients, 12.7% developed infections, a rate that falls within the expected range. Infections were more frequent among patients with LOS ≥ 7 days (42.9% vs. 1.1%). A three-parameter model (qSOFA score ≥ 2, age ≥ 65, and MDW ≥ 20) demonstrated good performance for new infection (AUC: 0.834) and LOS ≥ 7 days (AUC: 0.714). Adding CRP ≥ 2 mg/dL improved predictions for infection (AUC: 0.909) and LOS ≥ 7 days (AUC: 0.798), outperforming the C-ACS score (AUC: 0.807) and SHR (AUC: 0.784).

---

**Data availability statement:** Data cannot be shared publicly due to legal restrictions imposed by the government of Taiwan under the "Personal Information Protection Act." The data that support the findings of this study are available from the Office of Data Science (https://ods.tmu.edu.tw/) and the Office of Human Research (https://ohr.tmu.edu.tw/) at Taipei Medical University. Data access is restricted due to ethical and legal considerations, and requires a formal application and approval. Requests for data access can be submitted through the official websites of these offices by researchers who meet the criteria for access to confidential data.

**Funding:** This study was funded by Taipei Medical University (reference number: TMU111-AE1-B07, TMU112-AE1-B25), Taipei, Taiwan and by the National Science and Technology Council, Taipei, Taiwan (grant number: NSTC113-2314-B-038-051).

**Competing interests:** All authors declared no conflict of interest.

## Conclusion

MDW emerges as a promising biomarker for assessing the risks of infection and prolonged hospital LOS in patients with AMI. Its early use may facilitate timely clinical interventions, including earlier initiation of antibiotic therapy.

## Introduction

Newly diagnosed infection is a significant complication among patients undergoing primary percutaneous coronary intervention (PCI) for both ST-segment elevation myocardial infarction (STEMI) and non-ST-segment elevation myocardial infarction (NSTEMI) [1–5]. Studies indicate that 2.4% to 16.6% of patients with these conditions develop infections [2–5], with research on an octogenarian population revealing an approximate 30.5% infection rate among patients with acute myocardial infarction (AMI) [6]. While relatively uncommon, infections can significantly disrupt the treatment trajectory of STEMI and NSTEMI patients, leading to a prolonged hospital length of stay (LOS) and a fivefold increase in mortality risk [3,5,6]. Notably, patients who develop infections after STEMI and NSTEMI tend to experience longer LOS, typically ranging from 7 to 20 days, compared to those without infections, whose LOS typically spans 5–7 days [2,7].

Monocyte distribution width (MDW) stands as a novel parameter derived from routine complete blood count (CBC) analysis [8]. Utilizing laser-based technology, advanced hematology analyzer such as the Beckman Coulter UniCel DxH 800 and 900 can detect various degrees of morphological changes in monocytes [9,10]. An elevated MDW signifies activation of innate monocytes in response to microbial attacks [8,11–13]. This reflects two key pathophysiological processes: (1) pathogen-induced monocyte activation, characterized by cytoplasmic and nuclear changes; (2) increased volumetric dispersion of monocytes, a hallmark of early immune engagement [8]. As first-line responders, monocytes undergo size and shape changes proportional to the magnitude of infectious exposure —be it bacterial, viral, or fungal [9]. These morphological transformations arise from inflammatory signaling cascades and serve as an immediate defense mechanism. Clinically, MDW offers valuable adjunctive information beyond conventional leukocytosis, enhancing early infection detection in various acute settings.

Since 2020, several cardiac risk scoring systems have been explored for their potential to predict newly acquired infections during hospitalization, such as the Canada Acute Coronary Syndrome Risk (C-ACS) score [1,14,15], the stress hyperglycemia ratio (SHR) [16,17], the age –creatinine–ejection fraction score (or the age, estimated glomerular filtration rate, and ejection fraction score) [15], the Global Registry of Acute Coronary Events score [15], the Mehran score [15], and a 24-point risk scoring system [18] using extensive data on demographic characteristics, physiological measures, medication history, and laboratory tests.

While these models offer varying degrees of predictive value, they were primarily designed to assess cardiovascular risk rather than to detect early systemic infection.

Among these, only the C-ACS score [1,14,15] and SHR [16,17] have been validated for infection prediction in two or more independent studies. The C-ACS risk score—comprised of age ≥ 75 years, Killip class > 1, systolic blood pressure < 100 mmHg, heart rate > 100 beats per minute—has shown associations with infection risk when scores are ≥ 2 [1]. Nonetheless, its primary application remains short-term mortality prediction in patients with AMI, rather than infection surveillance. Similarly, the SHR model is susceptible to variability due to fluctuating glucose levels during acute stress responses in AMI and is confounded by inconsistencies in the timing of glucose measurements relative to infection onset. Although both C-ACS and SHR have been repurposed to infection risk stratification, neither incorporates immunologic or inflammatory biomarkers—limiting their sensitivity for early detection.

MDW, by contrast, has demonstrated utility as an early infection across different age groups, including children aged 6 years and above [12], adults from the general population [13], and older adults aged 60 years and above [11]. When combined with systemic inflammatory response syndrome (SIRS) and Quick Sequential Organ Failure Assessment (qSOFA) scores [12,13], MDW can significantly enhance the early detection of sepsis in emergency departments (EDs). To date, however, no study has specifically examined the efficacy of MDW for the early identification of infection in patients with AMI undergoing primary PCI representing a critical gap in the literature. Our primary goal was to determine the correlation between MDW and post-PCI infection in patients with AMI and the optimal cutoff value of MDW for infection detection. Our secondary objective was to compare the diagnostic performance of our MDW models with that of C-ACS and SHR models.

## Methods

### Study design and participants

The retrospective observational study consecutively included all adult patients aged 20 years and above diagnosed with AMI and underwent PCI at the ED of Taipei Medical University Hospital, a tertiary teaching hospital, between January 1, 2020, and September 30, 2021. Our study was conformed to the Declaration of Helsinki, and the written informed consent was waived due to de-linked and de-identified data were used. The study was approved by the Joint Institutional Review Board of Taipei Medical University (reference number: N202305105). The data were accessed for research purposes on July 10, 2023 using the TED-ED system in Taipei Medical University Hospital. Diagnosis of AMI was based on clinical evidence of acute myocardial ischemia, with or without changes in cardiac troponin levels, meeting at least one of the following criteria: (1) symptoms of myocardial ischemia, (2) new ischemic changes on electrocardiography, (3) presence of pathological Q waves on electrocardiography, (4) imaging-based evidence of new loss of viable myocardium or new regional wall motion abnormality in a pattern consistent with ischemic etiology, or (5) presence of a coronary thrombus on angiography. Patients with unstable angina, out-of-hospital cardiac arrest, those not undergoing PCI treatment, and those lacking MDW data at the ED were excluded from the study.

### Baseline characteristics and variable collection

For enrolled participants, demographic data including age, sex, and physiological parameters such as body temperature, heart rate (HR), respiratory rate, blood pressure upon ED arrival were collected. Medial comorbidities (hypertension, diabetes mellitus, prior coronary artery disease, stroke, and malignancy) and the clinical course during hospitalization were recorded. Venipuncture for peripheral blood sampling was conducted within 30 minutes of ED arrival, obtaining CBC analysis (including MDW), cardiac enzymes, and C-reactive protein (CRP) levels. Using Sepsis-3 criteria [19], the qSOFA score (altered mental status, respiratory rate of ≥22 breaths/min, and systolic blood pressure of ≤100 mmHg) was assessed in ED. Sepsis was diagnosed in patients when evidence of infection was accompanied by an increase in the SOFA score of ≥2.

### Biomarkers measured

CBC and MDW were measured using a multiparameter hematology analyzer (UniCel DxH 900; Beckman Coulter, Brea, CA, USA). Using this analyzer, monocytes were identified depending on their cell volume, high-frequency conductivity,

and laser light scattering capacity [10]. Blood samples for CBC and MDW analysis were collected in tubes containing $K_3$ EDTA as the anticoagulant. Cardiac enzymes and inflammatory markers, including C-reactive protein (CRP), were analyzed using a clinical chemistry analyzer (cobas c502 and c702; Roche Diagnostics, Taipei, Taiwan) with manufacturer-supplied reagents (Roche Diagnostics). For these tests, blood samples were drawn into serum-separator tubes, and all results were reported in International System of Units.

## Outcome measures

The primary outcome was on identifying newly acquired infection during hospitalization in patients with acute coronary syndrome treated with primary PCI. Post-PCI infection was determined based on each patient's symptoms and laboratory test results, as well as the subsequent administration of antibiotic treatment (confirmed by authors S-F Lin, H-A Lin). Disagreements were resolved though consultation with another author, S-K Hou, to reach consensus. The secondary outcome assessed LOS ≥ 7 days, defined as the duration from ED admission to discharge. All patients were followed up until their discharge from the hospital.

## Statistical analysis

Continuous variables were presented as mean ± standard deviation or median with interquartile range, while discrete variables were presented as proportions. Participants were categorized based on (1) presence of new infection and (2) LOS ≥ 7 days according to the outcome measures Comparisons between groups were conducted using the Student's t-test or the Mann–Whitney U test for continuous variables. Categorical variables were compared using the Pearson's chi-squared test or Fisher's exact test, as appropriate. Univariable and multivariable logistic regression models were utilized to determine the odds ratios (ORs) and 95% CIs. The dependent variables were (1) the presence of new infections and (2) LOS ≥ 7 days. Candidate variables for multivariable analysis were selected based on clinical relevance and a univariable $P$ value threshold of less than 0.05. Independent variables included patient characteristics such as age, sex, chest pain symptoms, dyspnea, syncope, vital signs, blood pressure, qSOFA score, laboratory biomarkers of cardiac enzymes, white blood cell counts, neutrophil-to-lymphocyte ratio (NLR), MDW, and C-reactive protein (CRP) level. Youden's index (cutoff point with maximum sensitivity + specificity − 1) was employed to identify the optimal threshold. Receiver operating characteristic (ROC) curve analysis was conducted to determine the area under the curve (AUC) and 95% Cis. AUC values of 0.9–1.0 were defined as excellent, 0.8–0.9 as very good, 0.7–0.8 as good, 0.6–0.7 as satisfactory, and <0.6 as unsatisfactory [20]. The goodness-of-fit for multivariable logistic regression was assessed using the Hosmer–Lemeshow test, with a model was considered to have good fitness when $P > 0.05$. Subgroup analysis was performed the STEMI and NSTEMI patients. All statistical analyses were conducted using SAS software version 9.4 (SAS Institute, Cary, NC, USA), with $P$ values less than 0.05 being considered statistically significant.

**Comparative analysis of models for predicting newly acquired infection.** We compared our new models with the C-ACS score and SHR models. The C-ACS score, ranging from 0 to 4, assigns 1 point for each of the following parameters: (1) age ≥ 75 years, (2) Killip classification of >1, (3) systolic blood pressure ≤ 100 mmHg, and (4) HR > 90 beats/min. The SHR was calculated using the formula: admission blood glucose (mg/dL)/(28.7 × HbA1c (%) − 46.7). We assessed the AUC of the ROC curves with our models using an integrated discrimination improvement (IDI) test, and evaluated the goodness-of-fit for each model using the Hosmer–Lemeshow test. To account for multiple comparisons, we applied Bonferroni correction by adjusting the significance threshold using the formula α/m, where α = 0.05 and m was the total number of comparisons.

**Sensitivity analysis.** Sensitivity Analysis was conducted by replacing the qSOFA score with SIRS score in the multivariable logistic regression models. A SIRS score [17] was assessed based on the following parameters: (1) a body temperature of >38°C or <36°C, (2) a HR of >90 beats/min, (3) a respiratory rate of >20 breaths/min, and (4) a white blood cell count of <4000 cells/μL or >12000 cells/μL.

## Results

### Participant characteristics

A total of 252 adult patients diagnosed with AMI (aged 20 years and above) and undergoing primary PCI were enrolled between January 1, 2020, and September 30, 2021. Among them, 138 had NSTEMI and 114 had STEMI upon admission to the ED. The flow diagram was shown (S1 Fig in S1 File). The mean age of the patients was $62.4 \pm 14.2$ years. Venipuncture was performed within 30 minutes of arrival at the ED, and CBC and MDW measurements were performed within 60 minutes. Among all patients, the mean white blood cell (WBC) counts was $18.1 \pm 2.4 \times 10^9$/L (reference range: $4.0–10.0 \times 10^9$/L), the mean absolute neutrophil count was $7.48 \pm 3.56 \times 10^9$/L (reference range: $1.8–7.7 \times 10^9$/L), the mean lymphocyte count was $2.44 \pm 1.62 \times 10^9$/L (reference range: $1.0–3.2 \times 10^9$/L), the mean monocyte count was $0.77 \pm 0.32 \times 10^9$/L (reference range: $0.2–0.8 \times 10^9$/L), the mean NLR was $4.6 \pm 4.3$, and the MDW was $18.1 \pm 2.4$ unit (reference range: < 20 units). Regarding the clinical chemistry tests, CRP reference range was < 5 mg/L, the mean creatine kinase was $445.1 \pm 878.1$ unit/L (reference: 40–200 U/L), creatine kinase MB subunit was $47.3 \pm 65.4$ unit/L (reference: < 25 U/L), and median Troponin-T was 83 ng/L (interquartile range of 31–245; reference range: < 14 ng/L).

### Incidence of newly diagnosed infection

During hospitalization, 12.7% (32 out of 252) of the patients were diagnosed with newly acquired infection during hospitalization, among which 43.8% (14 of 32) developed sepsis. The most common types of infection were pneumonia (n = 24), urinary tract infection (n = 6), and intra-abdominal infection (n = 2). The majority of patients had newly diagnosed infection within 3 days of hospitalization (S2 Fig in S1 File). S1 Table in S1 File presents the detailed characteristics of the 32 patients with infection. To further explore the temporal relationship between MDW levels and infection onset, we performed an exploratory time-to-infection analysis focusing on the first 3 days post-admission. As shown in S3 Fig in S1 File, a box-and-whisker plot displays MDW values stratified by the day of infection diagnosis. Notably, the median MDW on day 0—the day infection was diagnosed—was above 20 and higher than values observed on days 1 and 2. This pattern supports the clinical utility of MDW ≥ 20 as an early diagnostic threshold.

### Comparison between patients with and without infection

Patients with AMI and infection were significantly older ($73.3 \pm 14.2$ vs. $60.8 \pm 13.5$ years, $P < 0.0001$) compared to those without infection (Table 1). In addition, a higher proportion of infected patients exhibited SIRS score ≥2 (53.1% vs. 18.6%, $P < 0.0001$) and a qSOFA score ≥2 (21.9% vs. 3.2%, $P < 0.0001$) in the ED. Moreover, patients with infection had longer LOS in the hospital ($22.9 \pm 18.2$ vs. $5.1 \pm 3.7$ days, $P < 0.0001$). Elevated levels of inflammatory marker such as white blood cell counts ($13.1 \pm 3.6$ vs. $10.6 \pm 3.8$ per $10^3$ cells/µL, $P = 0.0006$), CRP levels ($6.3 \pm 7.8$ vs. $2.6 \pm 4.4$ mg/dL, $P = 0.0278$), and NLR ($8.1 \pm 7.9$ vs. $4.1 \pm 3.3$, $P < 0.0001$) were observed in infected patients, along with a higher proportion having an MDW of ≥20 (56.3% vs. 10.0%, $P < 0.0001$).

### Characteristics of patients with prolonged LOS

Among patients with LOS ≥ 7 days, 42.9% acquired new infections, whereas no sepsis was observed in patients with LOS < 7 days (Table 2). Similar to those with newly diagnosed infection, a higher proportion of patients with prolonged LOS ≥ 7 days exhibited SIRS score ≥ 2 (42.9% vs. 15.4%, $P < 0.0001$) and qSOFA score ≥ 2 (12.9% vs. 2.8%, $P < 0.0001$). In addition, they showed elevated levels of WBC count ($11.8 \pm 3.6$ vs. $10.6 \pm 3.9$ per $10^3$ cells/µL, $P = 0.0289$), CRP level ($5.2 \pm 6.9$ vs. $1.8 \pm 3.2$ mg/dL, $P = 0.0289$), NLR ($6.0 \pm 6.2$ vs. $4.1 \pm 3.2$, $P = 0.0159$), and MDW ≥ 20 (31.4% vs. 9.9%, $P < 0.0001$).

**Table 1. Characteristics of patients with acute myocardial infarction classified by newly diagnosed infection during hospitalization. (N = 252).**

| Characteristics (N = 252) | Classified by infection | | |
|---|---|---|---|
| | Without infection (N = 220) | With infection (N = 32) | P value |
| Age (years) | 60.8 ± 13.5 | 73.3 ± 14.2 | <0.0001* |
| Female sex | 38/220 (17.3%) | 10/32 (31.3%) | 0.0599 |
| BMI (kg/m²) | 26.1 ± 3.9 | 26.0 ± 3.5 | 0.9554 |
| Vital signs at ED visits | | | |
| Body temperature (°C) | 36.5 ± 0.4 | 36.7 ± 0.8 | 0.2450 |
| Heart rate (beats/min) | 87.5 ± 24.1 | 98.7 ± 28.5 | 0.0175* |
| Respiratory rate (breaths/min) | 18.4 ± 2.7 | 20.8 ± 4.4 | 0.0040* |
| SBP (mmHg) | 140.0 ± 37.6 | 136.3 ± 39.3 | 0.6050 |
| DBP (mmHg) | 83.7 ± 24.5 | 77.2 ± 25.1 | 0.1601 |
| MAP (mmHg) | 102.5 ± 27.8 | 96.9 ± 28.4 | 0.2902 |
| SIRS score | 0.8 ± 0.8 | 1.6 ± 1.2 | 0.0011* |
| SIRS score ≥ 2 (%) | 41/220 (18.6%) | 17/32 (53.1%) | <0.0001* |
| qSOFA score | 0.2 ± 0.5 | 0.8 ± 0.8 | 0.0002* |
| qSOFA score ≥ 2 (%) | 7/220 (3.2%) | 7/32 (21.9%) | <0.0001* |
| Medical comorbidity (%) | | | |
| Hypertension | 111/220 (50.5%) | 18/32 (56.3%) | 0.5400 |
| Diabetes mellitus | 63/220 (28.6%) | 10/32 (31.3%) | 0.7607 |
| Prior CAD | 69/220 (31.4%) | 12/32 (37.5%) | 0.4874 |
| Previous stroke | 3/220 (1.4%) | 2/32 (6.3%) | 0.1224 |
| Malignancy | 2/220 (0.9%) | 2/32 (6.3%) | 0.0797 |
| Clinical course | | | |
| Length of stay (day) | 5.1 ± 3.7 | 22.9 ± 18.2 | <0.0001* |
| Sepsis (%) | 0/220 (0%) | 14/32 (43.8%) | <0.0001* |
| ICU admission | 203/220 (92.3%) | 32/32 (100.0%) | 0.1034 |
| Clinical diagnosis | | | — |
| NSTEMI | 120/220 (54.6%) | 18/32 (56.3%) | |
| STEMI | 100/220 (45.5%) | 14/32 (43.8%) | |
| Cardiac enzymes | | | |
| CK (unit/L) | 461.6 ± 926.1 | 332.2 ± 415.0 | 0.1831 |
| CK-MB (unit/L) | 46.8 ± 68.5 | 50.7 ± 38.3 | 0.6380 |
| Median Troponin-T (ng/L) | 76 (30–227) | 112 (50–550) | 0.0721 |
| Inflammatory markers | | | |
| WBC (×10⁹/L) | 10.6 ± 3.8 | 13.1 ± 3.6 | 0.0006* |
| Monocyte count (×10⁹/L) | 0.770 ± 0.310 | 0.817 ± 0.365 | 0.4307 |
| Median CRP (mg/dL)† | 0.6 (0.2–2.5) | 3.6 (1.5–11.8) | 0.0002* |
| Mean MDW (unit) | 17.7 ± 1.8 | 20.8 ± 3.7 | <0.0001* |
| Median MDW (unit) | 17.5 (16.4–18.6) | 20.9 (17.9–23.2) | <0.0001* |
| MDW ≥ 20 (unit) | 22/220 (10.0%) | 18/32 (56.3%) | <0.0001* |
| Median NLR | 3.1 (1.8–5.6) | 5.6 (2.2–10.4) | 0.0070* |

Abbreviations: BMI, body mass index; BP, blood pressure; CAD, coronary artery disease; CK, creatine kinase; CRP, C-reactive protein; DBP, diastolic blood pressure; ED, emergency department; ICU, intensive care unit; MAP, mean arterial pressure; MDW, monocyte distribution width; NLR, neutrophil-to-lymphocyte ratio; NSTEMI, non-ST-segment elevation myocardial infarction; SBP, systolic blood pressure; SIRS, systemic inflammatory response syndrome; SOFA, Sequential Organ Failure Assessment; STEMI, ST-segment elevation myocardial infarction.

Student's t test and Pearson's chi-squared test were used to obtain P values.

*Statistically significant (P < 0.05).

†CRP levels were measured in only 107 patients in the ED.

Contiguous variables were expressed as mean ± standard deviation or median (interquartile range).

**Table 2. Characteristics of patients with acute myocardial infarction classified by length of stay more than 7 days uuring hospitalization. (*N* = 252).**

| Characteristics | Classified by length of stay (LOS) | | |
|---|---|---|---|
| | LOS < 7 days (*n* = 182) | LOS ≥ 7 days (*n* = 70) | *P* value |
| Age (years) | 59.8 ± 13.1 | 69.2 ± 14.7 | <0.0001* |
| Female sex | 31/182 (17.0%) | 17/70 (24.3%) | 0.1891 |
| Body mass index (kg/m²) | 26.2 ± 4.0 | 25.5 ± 3.1 | 0.2848 |
| Vital signs at ED visits | | | |
| Body temperature (°C) | 36.6 ± 0.4 | 36.6 ± 0.7 | 0.6634 |
| Heart rate (beats/min) | 85.8 ± 22.5 | 97.1 ± 28.9 | 0.0040* |
| Respiratory rate (/min) | 18.2 ± 2.2 | 19.9 ± 4.3 | 0.0020* |
| Systolic BP (mmHg) | 140.7 ± 35.7 | 136.5 ± 42.6 | 0.4216 |
| Diastolic BP (mmHg) | 84.6 ± 23.1 | 78.3 ± 27.8 | 0.0700 |
| MAP (mmHg) | 103.0 ± 26.3 | 97.7 ± 31.6 | 0.1539 |
| SIRS score ≥ 2 (%) | 28/182 (15.4%) | 30/70 (42.9%) | <0.0001* |
| qSOFA score ≥ 2 (%) | 5/182 (2.8%) | 9/70 (12.9%) | 0.0037* |
| Medical comorbidity (%) | | | |
| Hypertension | 88/182 (48.4%) | 41/70 (58.6%) | 0.1460 |
| Diabetes mellitus | 52/182 (28.6%) | 21/70 (30.0%) | 0.8228 |
| Prior CAD | 52/182 (28.6%) | 29/70 (41.4%) | 0.0503 |
| Previous stroke | 3/182 (1.7%) | 2/70 (2.9%) | 0.6196 |
| Malignancy | 2/182 (1.1%) | 2/79 (2.9%) | 0.3088 |
| Clinical course | | | |
| Length of stay (day) | 3.9 ± 1.2 | 16.4 ± 14.2 | <0.0001* |
| Sepsis (%) | 0/182 (0%) | 14/70 (20.0%) | <0.0001* |
| ICU admission | 167/182 (91.8%) | 68/70 (97.1%) | 0.1269 |
| Infection (%) | 2/182 (1.1%) | 30/70 (42.9%) | <0.0001* |
| Pneumonia | 1/182 (1.1%) | 23/70 (32.9%) | |
| Urinary tract infection | 1/182 (1.1%) | 5/70 (7.1%) | |
| Intra-abdominal infection | 0/182 (0%) | 2/70 (2.9%) | |
| Clinical diagnosis | | | 0.4511 |
| NSTEMI | 97/182 (53.3%) | 41/70 (58.6%) | |
| STEMI | 85/182 (46.7%) | 29/70 (41.4%) | |
| Cardiac enzymes | | | |
| CK (unit/L) | 431.7 ± 811.3 | 479.8 ± 1036.7 | 0.7276 |
| CK-MB (unit/L) | 47.6 ± 73.5 | 46.7 ± 37.6 | 0.8990 |
| Median Troponin-T (ng/L) | 74 (24–203) | 112 (43–554) | 0.009* |
| Inflammatory markers | | | |
| WBC (×10⁹/L) | 10.6 ± 3.9 | 11.8 ± 3.6 | 0.0289* |
| Monocyte count (×10⁹/L) | 0.760 ± 0.316 | 0.816 ± 0.320 | 0.2170 |
| Median CRP (mg/dl)† | 0.4 (0.2–2.1) | 2.5 (0.6–9.9) | 0.0002* |
| Mean MDW (unit) | 17.6 ± 1.9 | 19.2 ± 3.2 | 0.0002* |
| Median MDW (unit) | 17.4 (16.4–18.6) | 18.5 (17.0–21.1) | 0.0002* |
| MDW ≥ 20 (unit) | 18/182 (9.9%) | 22/70 (31.4%) | <0.0001* |
| Median NLR | 3.1 (1.8–5.5) | 4.2 (1.9–7.9) | 0.0738 |

Abbreviations: BMI, body mass index; BP, blood pressure; CAD, coronary artery disease; CK, creatine kinase; CRP, C-reactive protein; ED, emergency department; ICU, intensive care unit; MAP, mean arterial pressure; MDW, monocyte distribution width; NLR, neutrophil-to-lymphocyte

**Table 2.** (Continued)

ratio; SIRS, systemic inflammatory response syndrome; SOFA, Sequential Organ Failure Assessment; STEMI, ST-segment elevation myocardial infarction.

Student's t test and Pearson's chi-squared test were used to obtain *P* values.

*Statistically significant (*P*<0.05).

†CRP levels were measured in only 107 patients in the ED.

Contiguous variables were expressed as mean±standard deviation or median (interquartile range).

### Primary outcome measure: Predictors of newly diagnosed infection

The univariable analysis revealed several significant predictors of newly diagnosed infection (Table 3). Notably, age≥65 years (OR: 4.67, *P*=0.0003), HR≥90 beats/min (OR: 2.45, *P*=0.0214), respiratory rate ≥20 breaths/min (OR: 5.10, *P*<0.0001), SIRS score ≥2 (OR: 4.95, *P*<0.0001), qSOFA score ≥2 (OR: 8.52, *P*=0.0002), white blood cell count ≥10,500/µL (OR: 2.64, *P*=0.0165), NLR≥4 (OR: 2.59, *P*=0.0186), and MDW≥20 (OR: 11.57, *P*<0.0001) emerged as strong predictors. Further diagnostic performance detailed were provided in S2 Table in S1 File.

In the multivariable logistic regression model analysis (Table 4), incorporating variables such as the Quick Sequential Organ Failure Assessment (qSOFA) score, MDW, and factors associated with early infection, the full multivariable model yielding an AUC of 0.917 (95% CI, 0.867–0.968). The stability of the goodness-of-fit was confirmed by the Hosmer–Lemeshow test. Accordingly, we constructed two models: a three-parameter model (qSOFA score ≥2, age≥65 years, MDW≥20), and a four-parameter model (adding CRP≥2 mg/dL). Both models (Fig 1) achieved excellent AUC values for predicting newly diagnosed infection. The four-parameter model demonstrated an excellent AUC of 0.909 (95% CI: 0.860–0.958), with significant predictors including qSOFA score ≥2 (OR: 5.13, *P*=0.0288), age≥65 years (OR: 7.98, *P*=0.0003), CRP≥2 mg/dL (OR: 14.72, *P*<0.0001), and MDW≥20 (OR: 7.73, *P*=0.0001). The simplified three-parameter model demonstrated very good performance, with an AUC of 0.827 (95% CI: 0.751–0.902), with significant predictors of qSOFA score ≥2 (OR: 4.50, *P*=0.0454), age≥65 years (OR: 4.68, *P*=0.0015), and MDW≥20 (OR: 11.72, *P*<0.0001).

Additionally, for the multivariable logistic regression models treating MDW and CRP as continuous variables, both remained significant predictors of newly diagnosed infection. In the four-parameter model, MDW (OR: 1.67 per unit increase; *P*<0.0001) and CRP (OR: 1.17 per 1 mg/dL increase; *P*=0.0006) were independently associated with infection, yielding an AUC of 0.890 (95% CI, 0.827–0.952). In the three-parameter model, MDW (OR: 1.55 per unit increase; *P*<0.0001) remained a strong predictor, yielding an AUC of 0.843 (95% CI, 0.762–0.925). These findings support the robustness of MDW in both categorical and continuous formats for early infection detection in AMI patients.

### Secondary outcome measures: LOS≥7 days

S3 Table in S1 File presents the univariate analysis results for the correlation between the predictors of newly diagnosed infection and LOS≥7 days. Similar to the primary outcome measure, age≥65 years (OR: 3.24, *P*<0.0001), HR≥90 beats/min (OR: 2.58, *P*=0.0010), respiratory rate ≥20 breaths/min (OR: 3.42, *P*<0.0001), SIRS score ≥2 (OR: 4.13, *P*=0.0003), qSOFA score ≥2 (OR: 7.13, *P*=0.0007), NLR≥4 (OR: 2.14, *P*=0.0078), and MDW≥20 (OR: 11.57, *P*<0.0001) were significantly associated with LOS≥7 days. S3 and S4 Tables S1 File presents the univariate analysis results for the prediction of LOS≥7 days.

In the multivariable analysis (S5 Table in S1 File), the full multivariable model achieved good test quality for predicting LOS≥7 days, with AUC values of 0.798 (95% CI: 0.730–0.865). For the simplification, the three-parameter model with qSOFA score ≥2 (OR: 3.29, *P*=0.0658), age≥65 years (OR: 3.10, *P*=0.0002), and MDW≥20 (OR: 3.93, *P*=0.0004) still achieved satisfactory test quality with an AUC value of 0.714 (95% CI: 0.645–0.783).

**Table 3. Univariable analysis of newly diagnosed infection predictors. (N = 252).**

| Predictors (N = 252) | OR (95% CI) | P value |
|---|---|---|
| Age ≥ 65 years | 4.67 (2.01–10.88) | 0.0003* |
| Sex (male vs. female) | 0.45 (0.20–1.02) | 0.0565 |
| BMI (kg/m²) | 1.00 (0.88–1.13) | 0.9553 |
| Vital signs at ED visits | | |
| Body temperature (°C) | 1.92 (0.95–3.88) | 0.0712 |
| Heart rate (beats/min) | 1.02 (1.00–1.03) | 0.0199* |
| Heart rate ≥ 90 beats/min | 2.45 (1.14–5.27) | 0.0214* |
| Respiratory rate (breaths/min) | 1.22 (1.10–1.35) | 0.0001* |
| Respiratory rate ≥20 breaths/min | 5.10 (2.35–11.06) | <0.0001* |
| SBP (mmHg) | 1.00 (0.99–1.01) | 0.6036 |
| DBP (mmHg) | 0.99 (0.98–1.00) | 0.1607 |
| MAP (mmHg) | 0.99 (0.98–1.00) | 0.2900 |
| SIRS score (per unit of increase) | 2.26 (1.54–3.30) | <0.0001* |
| SIRS score ≥ 2 | 4.95 (2.29–10.72) | <0.0001* |
| qSOFA score (per unit of increase) | 3.58 (2.09–6.12) | <0.0001* |
| qSOFA score ≥ 2 | 8.52 (2.76–26.29) | 0.0002* |
| Medical comorbidity | | |
| Hypertension | 1.26 (0.60–2.66) | 0.5406 |
| Diabetes mellitus | 1.13 (0.51–2.53) | 0.7608 |
| Prior CAD | 1.31 (0.61–2.84) | 0.4882 |
| Previous stroke | 4.82 (0.77–30.05) | 0.0918 |
| Malignancy | 7.27 (0.99–53.52) | 0.0516 |
| Laboratory tests | | |
| CK (unit/L) | 1.00 (1.00–1.00) | 0.4447 |
| CK-MB (unit/L) | 1.00 (1.00–1.00) | 0.7538 |
| Troponin-T (ng/mL) | 0.92 (0.66–1.27) | 0.6050 |
| MDW (per unit increase) | 1.63 (1.37–1.94) | <0.0001* |
| MDW ≥ 20 | 11.57 (5.07–26.42) | <0.0001* |
| WBC (×10⁹/L) | 1.16 (1.06–1.26) | 0.0017* |
| WBC ≥ 10500 cells/μL | 2.64 (1.19–5.84) | 0.0165* |
| CRP† (per mg/dL increase) | 1.14 (1.06–1.24) | 0.0008* |
| CRP ≥ 2 mg/dL† | 13.61 (5.93–31.27) | <0.0001* |
| NLR | 1.17 (1.08–1.27) | 0.0001* |
| NLR ≥ 4 | 2.59 (1.17–5.73) | 0.0186* |

Abbreviations: BMI, body mass index; CAD, coronary artery disease; CK, creatine kinase; CRP, C-reactive protein; DBP, diastolic blood pressure; ED, emergency department; ICU, intensive care unit; MAP, mean arterial pressure; MDW, monocyte distribution width; NLR, neutrophil-to-lymphocyte ratio; SBP, systolic blood pressure; SIRS, systemic inflammatory response syndrome; SOFA, Sequential Organ Failure Assessment; OR, odds ratio.

Univariate logistic regression models were used to obtain P values.

*Statistically significant (P < 0.05).

†CRP levels were measured in only 107 patients in the ED.

## Comparative analysis of models

Although the C-ACS score (OR: 3.33 per increase of a score; 95% CI: 2.15–5.16, P < 0.0001; AUC: 0.807, 95% CI: 0.732–0.881) and the SHR (OR: 9.59 per one unit of increase; 95% CI: 3.92–23.53, P < 0.0001; AUC: 0.784, 95% CI: 0.685–0.882) exhibited satisfactory results for new infections (Fig 2), the IDI tests indicated that our four- and three-parameter

**Table 4. Multivariable analysis results of infection in patients with acute coronary syndrome (N = 252).**

| Characteristics | Full multivariable model | | Four-parameter model | | Three-parameter model | |
|---|---|---|---|---|---|---|
| | OR (95% CI) | *P* value | OR (95% CI) | *P* value | | |
| Models using inflammatory marker MDW and CRP as dichotomized variables | | | | | | |
| qSOFA score ≥ 2 | 2.39 (0.46–12.44) | 0.2995 | 5.13 (1.18–22.20) | 0.0288* | 4.50 (1.03–19.61) | 0.0454* |
| Age ≥ 65 years | 7.33 (2.21–24.28) | 0.0011* | 7.98 (2.56–24.86) | 0.0003* | 4.68 (1.81–12.09) | 0.0015* |
| MDW ≥ 20 | 7.19 (2.41–21.48) | 0.0004* | 7.73 (2.75–21.72) | 0.0001* | 11.72 (4.67–29.43) | <0.0001* |
| CRP ≥ 2 mg/dL | 15.20 (4.50–51.32) | <0.0001* | 14.72 (5.11–42.40) | <0.0001* | | |
| Heart rate ≥ 90 beats/min | 0.94 (0.33–2.70) | 0.9118 | | | | |
| Respiratory rate ≥ 20/min | 2.82 (0.94–8.45) | 0.0649 | | | | |
| NLR ≥ 4 | 0.65 (0.21–2.03) | 0.4617 | | | | |
| WBC ≥ 10500 cells/μL | 1.32 (0.42–4.10) | 0.6331 | | | | |
| Model fit | | | | | | |
| AUC (95% CI) | 0.917 (0.867–0.968) | | 0.909 (0.860–0.958) | | 0.827 (0.751–0.902) | |
| AIC | 126.38 | | 122.52 | | 147.52 | |
| Hosmer–Lemeshow test | 8.89 (df = 7) | 0.2610 | 2.13 (df = 4) | 0.7116 | 7.42 (df = 3) | 0.0597 |
| Models using inflammatory marker MDW and CRP as continuous variables | | | | | | |
| qSOFA score ≥ 2 | 3.78 (0.56–25.58) | 0.1736 | 7.68 (1.26–46.95) | 0.0274* | 3.77 (0.81–17.68) | 0.0920 |
| Age ≥ 65 years | 4.44 (1.25–15.80) | 0.0212* | 4.49 (1.36–14.79) | 0.0136* | 3.58 (1.39–9.18) | 0.0080* |
| MDW (per unit increase) | 1.58 (1.21–2.06) | 0.0008* | 1.67 (1.30–2.14) | <0.0001* | 1.55 (1.30–1.86) | <0.0001* |
| CRP (per mg/dL increase) | 1.19 (1.07–1.33) | 0.0012* | 1.17 (1.07–1.29) | 0.0006* | | |
| Heart rate ≥ 90 beats/min | 0.67 (0.19–2.43) | 0.5440 | | | | |
| Respiratory rate ≥20/min | 3.31 (0.87–12.62) | 0.0796 | | | | |
| NLR ≥ 4 | 1.20 (0.35–4.10) | 0.7753 | | | | |
| WBC ≥ 10500 cells/μL | 2.35 (0.65–8.51) | 0.1924 | | | | |
| Model fit | | | | | | |
| AUC (95% CI) | 0.904 (0.844–0.964) | | 0.890 (0.827–0.952) | | 0.843 (0.762–0.925) | |
| AIC | 94.65 | | 118.02 | | 146.54 | |
| Hosmer–Lemeshow test | 2.44 (df = 8) | 0.9644 | 11.14 (df = 8) | 0.1937 | 5.08 (df = 8) | 0.7486 |

Abbreviations: AUC, area under the curve; BT, body temperature; CRP, C-reactive protein; df, degrees of freedom; NLR, neutrophil-to-lymphocyte ratio; OR, odds ratio; RR, respiratory rate; SIRS, systemic inflammatory response syndrome.

Multivariable logistic regression models were used to obtain *P* values.

*Statistically significant (*P* < 0.05).† CRP levels were measured in only 107 patients in the ED.

exhibited higher diagnostic performance compared to the C-ACS and SHR models (S6 Table in S1 File). To account for multiple comparisons, we applied a Bonferroni correction, adjusting the significance threshold to 0.0125 (α = 0.05 divided by four comparisons). The comparisons remained statistically significant under this correction.

### Sensitivity analysis

In the sensitivity analysis (S7 Table in S1 File), the full multivariable maintained excellent discriminative ability, achieving an AUC of 0.909 (95% CI: 0.847–0.970) for predicting newly diagnosed infection (S3 Fig in S1 File). Both the four- and three-parameter models also demonstrated good test quality, with AUCs of 0.907 (95% CI, 0.848–0.967) and of 0.834 (95% CI: 0.752–0.916), respectively. For the secondary outcome of LOS ≥ 7 days (S5 Table in S1 File), the three-parameter model maintained satisfactory test quality, yielding an AUC value of 0.798 (95% CI: 0.727–0.868). Notably, in model comparison (S8 Table in S1 File), both the four- and three-parameter models outperformed the C-ACS and SHR models in detecting new infections (S4 Fig in S1 File).

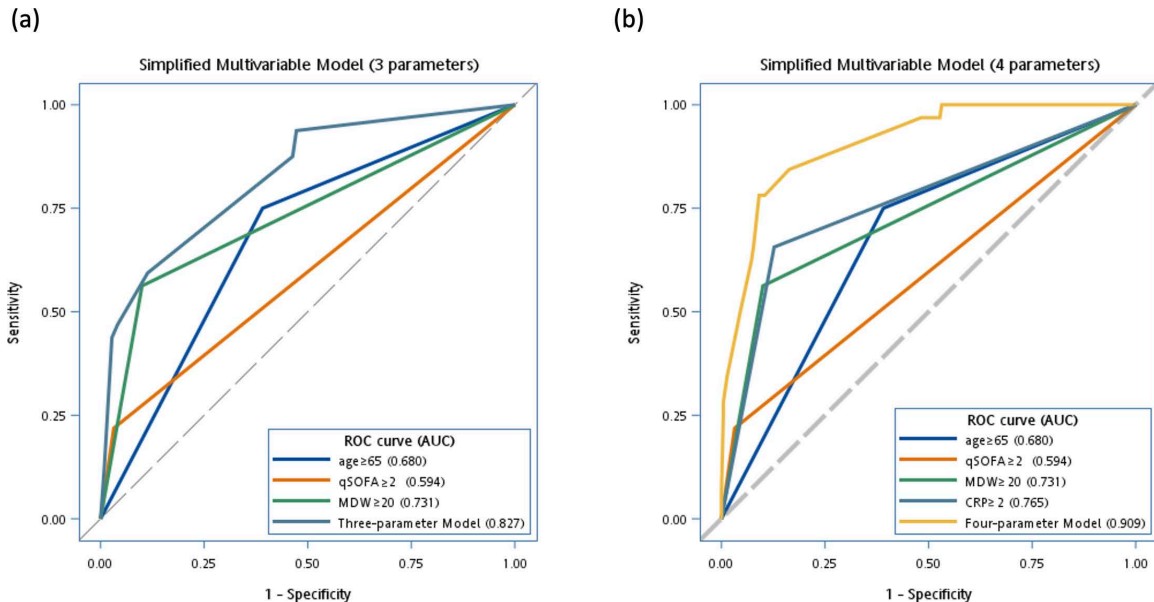

**Fig 1. Diagnostic performance of simplified models with three parameters (a) and four parameters (b).** While the three-parameter models included age ≥ 65 years, MDW ≥ 20, and qSOFA ≥ 2, the four-parameter models included the same three parameters in addition to CRP ≥ 2 mg/dL. (N = 252 for acute coronary syndrome patients were included for analysis).

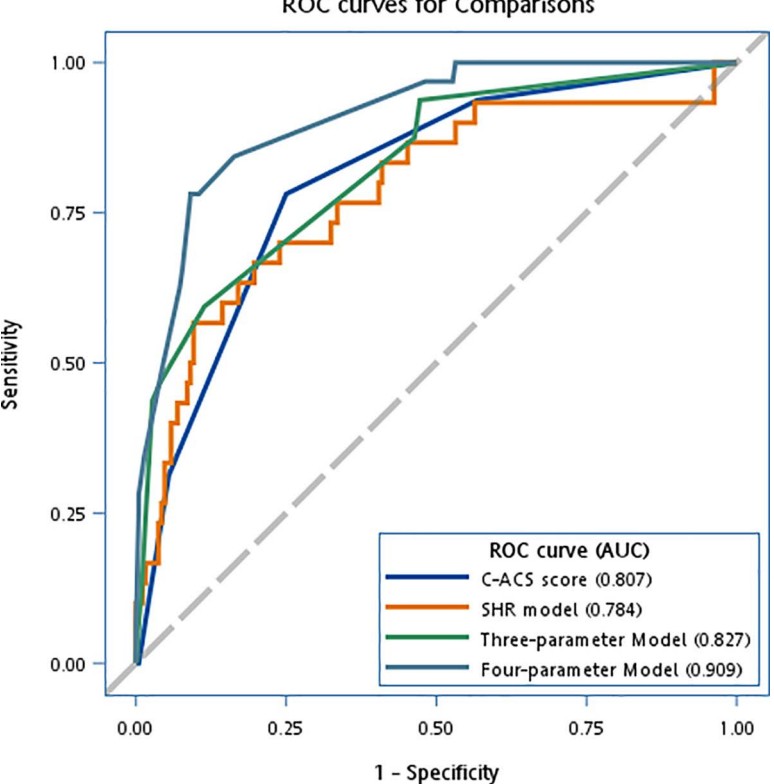

**Fig 2. Comparison of the diagnostic performance of the C-ACS and SHR models with our models for the prediction of infection during hospitalization.** (N = 252 for acute coronary syndrome patients were included for analysis).

## Subgroup for patients with STEMI and NSTEMI

Subgroup analyses for patients with STEMI and NSTEMI were shown (S9 Table in S1 File). Among patients with STEMI (n = 114), MDW ≥ 20 was not significantly associated with newly diagnosed infection in either the four-parameter model (OR: 2.39; *P* = 0.3620) or the three-parameter model (OR: 1.82; *P* = 0.4988). In contrast, for the NSTEMI subgroup (n = 138), MDW ≥ 20 remained a strong and consistent predictor for newly diagnosed infection across all model: full multivariable model (OR: 27.72, *P* = 0.0002), four-parameter model (OR: 29.76, *P* < 0.0001), and three-parameter model (OR: 50.05, *P* < 0.0001).

Nevertheless, caution is warranted in interpreting the results due to sample size consideration.

According to the widely accepted rule-of-thumb for binary outcome models, an events-per-variable (EPV) value of at least 20 is strongly recommended [21,22]. Based on the formula n = 100 + EPV × *I* (where *I* refers to the number of independent variables) [21,22], the minimum sample size required for developing three- and four-parameter model would be approximately 160 and 180, respectively.

## Discussion

In this study, 12.7% of patients with STEMI and NSTEMI who underwent primary PCI developed new infections during their hospital stay. New infections significantly contributed to prolonged LOS, particularly evident in patients with LOS exceeding 7 days (42.9% vs. 1.1% in those with LOS < 7 days). This underscores the clinical relevance of detecting and managing infections promptly in patients undergoing PCI. Our study provide validation for the utility of peripheral blood tests such as MDW in the ED setting to enhance the early detection of new infections. The multivariable analysis using qSOFA score-based models highlighted a threshold MDW of ≥20 was universally applicability for identifying patients with newly acquired infection. Notably, our simplified multivariable models, comprising only three parameters—qSOFA score ≥ 2, age ≥ 65 years, and MDW ≥ 20, exhibited superior performance (AUC: 0.8–0.9) for the identification of newly acquired infection than that of the C-ACS and SHR models (AUC: 0.7–0.8) in STEMI and NSTEMI patients.

This study is the first to examine the performance of MDW to detect infections on patients with STEMI and NSTEMI undergoing primary PCI. In this study, we discovered most of our patients acquired new infections within 3 days, thus confirming the importance of MDW measurements in the ED setting. In addition, the high prevalence of sepsis (20.0%) in the group with LOS ≥ 7 days explained the stronger effect of sepsis on LOS. Notably, our MDW threshold of ≥20 were compatible to previous studies conducted in ED settings to enhance the detection of sepsis [11,13,23–25]. The general characteristics observed in this study, including the prevalence of infection (12.7%) and sepsis (5.6%), the sites of infection, and the timing of diagnosis for newly acquired infection, are consistent with those reported in previous studies. An analysis in the APEX-AMI clinical trial in patients with STEMI revealed that 2.4% (138 of 5745) had newly acquired infection, predominantly single-site infection [2]. In a prospective cohort study, Piccaro et al. [3] reported that the median time to diagnosis of infection in STEMI patients was 3 days, and half of the infections were community-acquired. Santos et al. [7] discovered that 11.1% (126 of 1131) of patients with STEMI had nosocomial infection. In a study using the Florida surveillance database, Chen et al. [5] reported an increased prevalence (16.6%) of health-care-acquired infection in patients with STEMI, with approximately 4.0% (206 of 5216) of those with NSTEMI had new infection. Consistent to previous studies [2–5], most of our patients had the diagnoses of new infections within 3 days, with pneumonia being the most frequent, followed by urinary tract infection.

Our three-parameter models—qSOFA score ≥ 2, age ≥ 65 years, and MDW ≥ 20, had several advantages, and it outperformed both the C-ACS and SHR models with very good diagnostic performance (AUC: 0.8–0.9). While the C-ACS score includes vital signs such as SBP and HR, it lacked an inflammatory marker, limiting its ability for early identification of infection. In our cohort, 40.6% (13 of 32) of patients with STEMI and NSTEMI who had newly acquired infection did not exhibit fever during their hospital stay. These findings showed that MDW, indicative of the activation of innate immunity in response to microbial invasion, may predict potential infection, enhancing diagnostic accuracy beyond that achieved

by the C-ACS score. In addition, implementing SHR in ED settings presents challenges due to the nonroutine measurement of HbA1c. Contrastingly, MDW measurements, integrated into routine CBC testing, off a more convenient solution. MDW measurements, along with CBC, conducted on hematology analyzers have a shorter turnaround time compared to measurements of CRP, procalcitonin, and interleukin-6 with clinical chemistry analyzers [26–28]. Critically, MDW incurs no additional costs and requires no extra blood samples in routine CBC tests. Our four-parameter model demonstrated excellent predictive performance in EDs (AUC ≥ 0.9). Lastly, our decision to conduct a sensitivity analysis with SIRS score stemmed from the observed discrepancy in the definitions of sepsis between Sepsis-2 and Sepsis-3, along with the diverse phenotypes of sepsis [29,30].

Although comorbid conditions such as hypertension, diabetes mellitus, prior coronary artery disease, stroke, and malignancy are known to influence infection risk in hospitalized patients, none of these variables demonstrated a significant association with newly diagnosed infections in our univariable analysis (Table 2). Accordingly, they were not included in our multivariable models. This decision was guided by strict model-building strategies that incorporate only variables with statistically significant associations (p < 0.05) in univariable screening, especially when the number of outcome events is relatively limited. This approach helped to minimize the risk of overfitting while preserving model interpretability and clinical usability. Instead, we prioritized the use of simple, bedside-accessible clinical scoring system such as qSOFA and SIRS. These tools are routinely employed to evaluate systemic illness severity and have the added advantage of being immediately available in acute care settings—unlike many detailed comorbidity indices, which may not be fully accessible during early clinical decision-making.

This study has some limitations. First, this study was conducted in an ED where procalcitonin measurements were not available, precluding a direct comparison of procalcitonin and MDW's diagnostic performance in identifying infection. However, the latest Surviving Sepsis Campaign guidelines (2021) [31] advise against the combining procalcitonin measurements and clinical evaluation to determine the timing of antibiotic administration. In a meta-analysis of three randomized controlled trials [32–34] employing procalcitonin-guided protocols for antibiotic administration, no positive effects was found between the use of procalcitonin and mortality or LOS in hospitals [31]. Despite procalcitonin potentially aiding in identifying bacterial infection, it does not reduce mortality rates [35]. Additionally, procalcitonin is often associated with high costs, which may limit its availability in low-income countries [31]. As a part of routine CBC tests, MDW incurs no additional costs [8]. Notably, multiple studies have highlighted the crucial role of MDW in assessing the severity of sepsis and expediting decisions regarding antibiotic administration [11,12,24,25,36]. Second, this study had a retrospective cohort observational design, which presumably resulted in a residual confounding effect. The physicians included in this study were not blinded to the MDW values, which could have introduced a degree of observer bias. Addressing this limitation would ideally require a double-blind, randomized controlled trial, although such designs may pose ethical challenges in urgent care settings. Third, although we conducted subgroup analyses stratified by AMI subtype (STEMI and NSTEMI), the relatively small sample sizes in each group did not meet the recommended events-per-variable (EPV) threshold for robust multivariable modeling. While MDW ≥ 20 was a consistent predictor of infection in NSTEMI patients, its predictive value was not significant among STEMI patients—likely due to insufficient power rather than true biological difference. As such, these subgroup findings should be interpreted with caution, and future studies with larger samples are warranted to validate model performance across AMI subtypes.

Lastly, although MDW demonstrated strong diagnostic potential, our models have not yet been externally validated. MDW reflects the variability in monocyte size, which increases during infections due to activation and morphological transformation. These changes result in greater size dispersion, which is quantitatively captured as elevated MDW. This process is driven by pro-inflammatory cytokines such as IL-6 and TNF-α, which not only stimulate monocyte activation but also upregulate acute-phase reactants like CRP [37]. Therefore, MDW complements traditional systemic inflammatory markers such as CRP and WBC count by providing an early, cell-based reflection of innate immune activation. Unlike CRP, which may take hours to rise, MDW changes can occur rapidly, enhancing its value for early detection of infection.

Prospective, multicenter studies are needed to confirm the reproducibility and generalizability of MDW-based prediction tools in broader AMI populations. Furthermore, randomized controlled trials evaluating MDW-guided clinical strategies—such as earlier initiation of antibiotics or enhanced infection surveillance—could clarify its potential role in optimizing infection management following PCI.

In conclusion, our simplified models—comprising (1) qSOFA score ≥2, (2) age ≥ 65 years, and (3) MDW ≥ 20, with or without the fourth parameter of CRP ≥ 2 mg/dL—provide a straightforward approach and efficient strategy for early infection detection in AMI patients undergoing PCI in the ED. These easily remembered parameters are universally applicable, aligning with both Sepsis-2 and Sepsis-3 frameworks and are easy to implement in real-world clinical settings. Timely recognition of infection using MDW-based models holds the potential to significantly improve patient outcomes by facilitating earlier antibiotic administration, reducing diagnostic uncertainty, minimizing delays in clinical decision-making, and potentially shortening hospital length of stay. These benefits may help improve hospital efficiency and reduce overall healthcare costs. However, before widespread clinical adoption, further validation in larger, prospective, multicenter studies is essential to confirm the robustness, generalizability, and real-world utility of these models across diverse healthcare settings.

## Supporting information

**S1 File.** S1 Table. Characteristics of patients with acute myocardial infarction and newly diagnosed infections. S2 Table. Diagnostic performance of predictors for newly diagnosed infections. S3 Table. Univariable analysis results of length of stay ≥ 7 days. S4 Table. Diagnostic performance of the prediction of length of stay ≥ 7 days. S5 Table. Multivariable analysis results of length of stay ≥ 7 days (including the sensitivity analysis by replacing qSOFA with SIRS score). S6 Table. Comparisons for model for predicting newly diagnosed infection. S7 Table. Sensitivity analysis: multivariable analysis for prediction of new infection in patients with acute coronary syndrome (N = 252). S8 Table. Sensitivity analysis: comparisons for model for predicting newly diagnosed infection. S9 Table. Subgroup analysis for the results of newly diagnosed infection (N = 252). S1 Fig. The study flow diagram. N = 310 for acute coronary syndrome patients were screened for enrollment. S2 Fig. Timing of newly diagnosed infections during hospitalization. N = 32 for acute coronary syndrome patients treated after PCI with infection. S3 Fig. Distribution of Monocyte Distribution Width (MDW) by Infection Onset. This box-and-whisker plot displays the distribution of MDW values across different days of infection onset, with individual data points overlaid as jittered dots. Notably, the median MDW on day 0—the day of newly diagnosed infection—was above 20, and higher than that observed on subsequent days (days 1 and 2). This finding supports our predictive model, which identified MDW ≥ 20 as a significant threshold for newly diagnosed infection, reinforcing MDW's role as an early biomarker in acute infection settings. S4 Fig. Diagnostic performance of simplified models with three parameters (a) and four parameters (b). While the three-parameter models included age ≥ 65 years, MDW ≥ 20, and SIRS ≥ 2, the four-parameter models included the same three parameters in addition to CRP ≥ 2 mg/dL. S5 Fig. Sensitivity analysis: comparison of the diagnostic performance of the C-ACS and SHR models with our models for the prediction of infection during hospitalization. The area under the curve (AUC) was 0.807 for the C-ACS score, 0.784 for the SHR model, 0.834 for our three-parameter model, and 0.907 for our four-parameter model.
(DOCX)

## Author contributions

**Conceptualization:** Hui-An Lin, Peter C Hou, Hung-Wei Tsai, Sen-Kuang Hou.

**Data curation:** Sheng-Feng Lin, Hung-Wei Tsai, Sen-Kuang Hou.

**Formal analysis:** Sheng-Feng Lin, Hung-Wei Tsai, Sen-Kuang Hou.

**Funding acquisition:** Sheng-Feng Lin, Sen-Kuang Hou.

**Investigation:** Sheng-Feng Lin, Hui-An Lin.

**Methodology:** Sheng-Feng Lin.

**Software:** Sheng-Feng Lin, Sen-Kuang Hou.

**Supervision:** Peter C Hou, Sen-Kuang Hou.

**Validation:** Sheng-Feng Lin, Sen-Kuang Hou.

**Writing – original draft:** Sheng-Feng Lin, Sen-Kuang Hou.

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
