## [Decision Letter · Decision Letter 0]

7 Apr 2025

PONE-D-25-04404Monocyte Distribution Width Enhances the Detection of Infection in Patients After Primary Percutaneous Coronary InterventionPLOS ONE

Dear Dr. Hou,

Thank you for submitting your manuscript to PLOS ONE. After careful consideration, we feel that it has merit but does not fully meet PLOS ONE’s publication criteria as it currently stands. Therefore, we invite you to submit a revised version of the manuscript that addresses the points raised during the review process.

**ACADEMIC EDITOR** : All issues raised by expert reviewers are required. 

We look forward to receiving your revised manuscript.

Kind regards,

Vincenzo Lionetti, M.D., PhD

Academic Editor

PLOS ONE

“This study was funded by Taipei Medical University (reference number: TMU111-AE1-B07, TMU112-AE1-B25), Taipei, Taiwan and by the National Science and Technology Council, Taipei, Taiwan (grant number: 113-2314-B-038-051-).”

3. In the online submission form, you indicated that your data is available only on request from a third party. Please note that your Data Availability Statement is currently missing the contact details for the third party, such as an email address or a link to where data requests can be made. Please update your statement with the missing information.

Reviewers' comments:

Reviewer's Responses to Questions

**Comments to the Author**

1. Is the manuscript technically sound, and do the data support the conclusions?

Reviewer #1: Yes

Reviewer #2: Yes

2. Has the statistical analysis been performed appropriately and rigorously? 

Reviewer #1: Yes

Reviewer #2: Yes

3. Have the authors made all data underlying the findings in their manuscript fully available?

Reviewer #1: Yes

Reviewer #2: Yes

4. Is the manuscript presented in an intelligible fashion and written in standard English?

Reviewer #1: Yes

Reviewer #2: Yes

5. Review Comments to the Author

Reviewer #1: The manuscript investigates the utility of monocyte distribution width (MDW) as an early biomarker for detecting newly acquired infections and predicting prolonged hospital length of stay (LOS) in acute myocardial infarction (AMI) patients undergoing primary percutaneous coronary intervention (PCI). The study is timely, as infections post-PCI are associated with increased morbidity and mortality. The authors propose simple, parsimonious multivariable models incorporating MDW along with clinical parameters (qSOFA and age, with optional CRP) and compare these with established scoring systems (C-ACS and SHR).

In the abstract, consider rephrasing “and to assessed MDW-based models” to “and to assess MDW-based models…” for grammatical clarity.

The abstract quotes infection rates ranging from 2.4% to 16.6% but later reports a 12.7% rate in this cohort. It might be helpful to briefly explain how these rates compare or whether the study cohort falls within expected ranges.

While the AUC values are provided, a brief statement on the clinical impact (e.g., potential to initiate early antibiotic therapy) would strengthen the abstract’s impact.

Introduction

Although several scoring systems (C-ACS, SHR, etc.) are mentioned, a more detailed critique of their limitations in infection detection could help underline the novelty and clinical relevance of using MDW.

A brief discussion on the pathophysiological basis for MDW elevation in infections (i.e., monocyte activation and morphological changes) would add depth.

Explicitly state the knowledge gap that this study addresses—for example, “To date, no study has evaluated the diagnostic performance of MDW for early infection detection in AMI patients post-PCI.”

Methods

While several clinical and laboratory parameters were assessed, provide more details on how candidate variables were selected for inclusion in the multivariable models.

The use of Youden’s index and ROC analysis is appropriate. However, it would be useful to mention if any corrections for multiple comparisons were applied when comparing several models.

Results

Ensure that all supplemental tables and figures referenced (e.g., Supplemental Figures 1–4) are clearly labeled, with adequate legends to make them self-explanatory.

Given that the patient population includes both STEMI and NSTEMI patients, consider reporting whether the performance of the MDW-based models differed between these subgroups.

Although dichotomized variables (e.g., MDW ≥20) are used, presenting the distribution of MDW as a continuous variable could provide additional context.

Given that most infections were diagnosed within 3 days, a brief analysis of the time course of infection onset in relation to MDW values might add further insights.

Discussion

Discuss potential confounding factors in greater detail—for instance, how comorbidities and variations in treatment protocols might influence both MDW and infection outcomes.

Offer specific recommendations for future research, such as prospective multicenter studies or randomized controlled trials to validate the MDW thresholds and models.

Expand upon the biological rationale for why MDW is elevated in infection and how it might interact with other markers of systemic inflammation.

Conclusions

Consider emphasizing how early detection using MDW-based models might improve patient outcomes (e.g., reduced LOS, timely antibiotic administration) and potentially lower healthcare costs.

Stress the need for further validation in larger, prospective, and multicenter studies before widespread clinical implementation.

Reviewer #2: The manuscript of Sheng-Feng et al. reports a retrospective observational study focusing on monocyte distribution width (MDW) as predictor of infection and prolonged hospital length of stay in 252 adult patients with AMI. The authors have to consider the following mostly minor comments:

1) On page 5 in the paragraph “Biomarkers measured” please specify the clinical chemistry analyzer used for the determination of cardiac enzymes parameters and express both them and inflammatory markers in the recommended International System of Units (SI);

2) Both in table 1 and in table 2 add absolute monocyte count among inflammatory markers; add “(units)” after MDW parameter; report CRP and NLR values as median and interquartile range;

3) On page 7, in the paragraph “ Participant characteristics” of the Results section, add hematochemical parameters measurements (complete and differential blood count, creatinine and other analytes measured) and the reference range adopted;

4) Please add to the references: Martinez-Iribarren A, Tejedor X, Sala Sanjaume A, Leis A, Dolade Botias M, Morales-Indiano C. Performance evaluation of the new hematology analyzer UniCel DxH 900. Int J Lab Hematol 2021;43:623-31 where DxH 900 analyzer showed good analytical performance in measuring MDW, the novel parameter for the early diagnosis of sepsis studied by Sheng-Feng et al.

6. PLOS authors have the option to publish the peer review history of their article (what does this mean? ). If published, this will include your full peer review and any attached files.

**Do you want your identity to be public for this peer review?** For information about this choice, including consent withdrawal, please see our Privacy Policy .

Reviewer #1: No

Reviewer #2: No

---

## [Author Response · Author response to Decision Letter 1]

16 Apr 2025

We sincerely thank the reviewers for their constructive and insightful comments. In response, we have revised the manuscript accordingly to address each of the points raised. All changes made to the original manuscript have been highlighted in red text. Our detailed responses to each comment are provided below:

Responses to reviewer #1:

Reviewer #1: The manuscript investigates the utility of monocyte distribution width (MDW) as an early biomarker for detecting newly acquired infections and predicting prolonged hospital length of stay (LOS) in acute myocardial infarction (AMI) patients undergoing primary percutaneous coronary intervention (PCI). The study is timely, as infections post-PCI are associated with increased morbidity and mortality. The authors propose simple, parsimonious multivariable models incorporating MDW along with clinical parameters (qSOFA and age, with optional CRP) and compare these with established scoring systems (C-ACS and SHR).

• We thank the reviewer for the thoughtful summary and positive evaluation of our work. We appreciate your recognition of the clinical relevance and timeliness of investigating MDW as an early biomarker for post-PCI infections and prolonged hospital stay in AMI patients. Our aim was to develop practical and easily applicable predictive models that can support early clinical decision-making. We are pleased that our approach—incorporating MDW with simple clinical parameters and comparing it with established scoring systems—was well received. We have carefully addressed all specific comments and suggestions provided in the subsequent sections.

1. In the abstract, consider rephrasing “and to assessed MDW-based models” to “and to assess MDW-based models…” for grammatical clarity.

The abstract quotes infection rates ranging from 2.4% to 16.6% but later reports a 12.7% rate in this cohort. It might be helpful to briefly explain how these rates compare or whether the study cohort falls within expected ranges.

While the AUC values are provided, a brief statement on the clinical impact (e.g., potential to initiate early antibiotic therapy) would strengthen the abstract’s impact.

• Thank you for this valuable feedback. We have made the following revisions to the abstract to improve clarity and relevance: (1) corrected the grammatical error by changing “to assessed” to “to assess”; (2) clarified that the observed 12.7% infection rate falls within the previously reported range (2.4% to 16.6%), thereby providing context for our findings; and (3) added a statement highlighting the clinical significance of our results, particularly the potential role of MDW in guiding early therapeutic decisions, such as timely initiation of antibiotics. These changes enhance the overall clarity and clinical impact of the abstract. (Please see our revised manuscript, page 2)

2. Introduction

Although several scoring systems (C-ACS, SHR, etc.) are mentioned, a more detailed critique of their limitations in infection detection could help underline the novelty and clinical relevance of using MDW.

A brief discussion on the pathophysiological basis for MDW elevation in infections (i.e., monocyte activation and morphological changes) would add depth.

Explicitly state the knowledge gap that this study addresses—for example, “To date, no study has evaluated the diagnostic performance of MDW for early infection detection in AMI patients post-PCI.”

• Thank you for this insightful comment. In response, we have revised the Introduction section to strengthen the rationale and novelty of our study by incorporating the following enhancements:

• 1. Critique of Existing Scoring Systems: We added a more focused discussion on the limitations of current scoring models such as the C-ACS score and SHR. Specifically, we noted that although both have been validated for infection risk stratification in acute myocardial infarction (AMI), they rely on generalized clinical parameters rather than immune or inflammatory biomarkers. The C-ACS score, while associated with infection risk when ≥2, was originally designed to predict short-term mortality and does not reflect early immunologic changes. Similarly, the SHR is influenced by stress-induced glycemic variability and inconsistencies in timing of glucose measurements, limiting its reliability. Neither model incorporates real-time immune activation markers, which constrains their sensitivity for early infection detection. (See page 4, Introduction)

• 2. Pathophysiological Basis of MDW: We elaborated on the underlying mechanism of MDW elevation in infections. MDW is a novel hematologic parameter derived from advanced laser-based analysis of monocyte morphology. It reflects monocyte activation and volumetric dispersion in response to microbial invasion—key hallmarks of early innate immune response. These morphological transformations result from inflammatory signaling and offer a dynamic biomarker that complements traditional leukocytosis. MDW thus provides valuable early diagnostic insights in acute care settings. (See page 1, Introduction)

• 3. Statement of Knowledge Gap: We explicitly stated the unmet need that this study addresses: to our knowledge, no prior research has specifically evaluated the diagnostic utility of MDW for early infection detection in AMI patients undergoing primary PCI. This represents a critical knowledge gap in the field. (See page 5, Introduction) We believe these revisions enhance the scientific framing and clinical relevance of our study and thank you again for helping us improve the manuscript.

3. Methods

While several clinical and laboratory parameters were assessed, provide more details on how candidate variables were selected for inclusion in the multivariable models.

The use of Youden’s index and ROC analysis is appropriate. However, it would be useful to mention if any corrections for multiple comparisons were applied when comparing several models.

• Thank you for your insightful comments. We have revised the Methods section to clarify variable selection and address concerns regarding multiple comparisons, as outlined below:

• 1.Candidate Variable Selection: We clarified the selection of variables for multivariable analysis was guided by both clinical relevance and statistical significance in univariable analyses (P < 0.05). The revised text now reads: “Candidate variables for multivariable analysis were selected based on clinical relevance and a univariable P value threshold of less than 0.05.” (Please see page 7, methods)

• 2. Correction for Multiple Comparisons: To address the potential inflation of Type I error due to multiple model comparisons, we applied a Bonferroni correction. This has been explicitly stated in the revised Methods section as follows: “To account for multiple comparisons, we applied Bonferroni correction by adjusting the significance threshold using the formula α/m, where α = 0.05 and m was the total number of comparisons.” (Please see page 8, methods)

• 3. Corresponding Results Section: In the Results section (Please see page 18), we have clarified the adjusted significance threshold and confirmed that our findings remained statistically significant after correction: “To account for multiple comparisons, we applied a Bonferroni correction, adjusting the significance threshold to 0.0125 (α = 0.05 divided by four comparisons). The comparisons remained statistically significant under this correction.” We believe these revisions improve the transparency and statistical rigor of our methodology, and we thank the reviewer for highlighting this important point.

4. Results

Ensure that all supplemental tables and figures referenced (e.g., Supplemental Figures 1–4) are clearly labeled, with adequate legends to make them self-explanatory.

Given that the patient population includes both STEMI and NSTEMI patients, consider reporting whether the performance of the MDW-based models differed between these subgroups.

Although dichotomized variables (e.g., MDW ≥20) are used, presenting the distribution of MDW as a continuous variable could provide additional context.

Given that most infections were diagnosed within 3 days, a brief analysis of the time course of infection onset in relation to MDW values might add further insights.

• Thank you for your constructive comments. We have made the following revisions to the Results section to address your suggestions:

• 1. Labeling of Supplemental Figures and Tables: We acknowledge the oversight in our initial submission. In the revised manuscript, all supplemental figures (Supplemental Figures 1–4) are now clearly labeled with comprehensive legends to ensure they are self-explanatory and interpretable. We have also corrected the labeling error in Table 2 for consistency.

• 2. Subgroup Analysis by AMI Type (STEMI vs. NSTEMI): To assess whether MDW-based model performance differed by AMI subtype, we conducted a detailed subgroup analysis, now presented in Supplemental Table 9. Among NSTEMI patients (n = 138), MDW ≥ 20 remained a strong predictor of newly diagnosed infection in all models:

• Full model: OR 27.72, P = 0.0002

• Four-parameter model: OR 29.76, P < 0.0001

• Three-parameter model: OR 50.05, P < 0.0001

• In contrast, among STEMI patients (n = 114), MDW ≥ 20 was not significantly associated with infection in either the:

• Four-parameter model: OR 2.39, P = 0.3620

• Three-parameter model: OR 1.82, P = 0.4988

• To ensure appropriate model development and minimize overfitting, we adhered to the guideline of ≥20 events per variable (EPV) for binary prediction models. Using the formula Minimum Sample Size = 100 + (EPV × number of predictors), we found that both subgroups were underpowered for fully reliable modeling (minimum of 160–180 subjects needed). These limitations are clearly addressed in the revised manuscript to encourage cautious interpretation of the subgroup findings (page 18-19, results; and page 22, discussion).

• 3. Presentation of MDW as a Continuous Variable: In addition to the dichotomized analysis (MDW ≥ 20), we performed multivariable logistic regression treating MDW and CRP as continuous variables. As now detailed in the Results (page 16–17), both variables remained significant:

• Four-parameter model: MDW (OR: 1.67 per unit increase, P < 0.0001), CRP (OR: 1.17 per 1 mg/dL increase, P = 0.0006), AUC = 0.890 (95% CI: 0.827–0.952)

• Three-parameter model: MDW (OR: 1.55 per unit increase, P < 0.0001), AUC = 0.843 (95% CI: 0.762–0.925)

• These results support the robustness of MDW as a predictive biomarker, whether used categorically or continuously. The new findings are presented in revised Table 4. (page 16, and the revised text, page 17)

• 4. Time Course of Infection and MDW Dynamics: We conducted an exploratory time-to-infection analysis focusing on the first 3 days post-admission, during which most infections were diagnosed. S3 Fig now displays a box-and-whisker plot of MDW values stratified by day of infection onset. Individual data points are overlaid as jittered dots. Notably, the median MDW on day 0 (day of infection diagnosis) was above 20 and higher than values observed on days 1 and 2. This finding supports the clinical utility of MDW ≥ 20 as a threshold for early infection detection and reinforces its role as a real-time biomarker of innate immune activation. (please see page 10)

5. Discussion

Discuss potential confounding factors in greater detail—for instance, how comorbidities and variations in treatment protocols might influence both MDW and infection outcomes.

Offer specific recommendations for future research, such as prospective multicenter studies or randomized controlled trials to validate the MDW thresholds and models.

Expand upon the biological rationale for why MDW is elevated in infection and how it might interact with other markers of systemic inflammation.

• Thank you for these thoughtful and constructive suggestions. We have addressed each of your points as follows:

• Confounding by Comorbidities and Treatment Variation: We agree that baseline comorbidities and variations in treatment protocols may act as potential confounders in evaluating the association between MDW and infection risk. To explore this, we performed univariable analyses of key comorbid conditions, including hypertension, diabetes mellitus, prior coronary artery disease (CAD), previous stroke, and malignancy. None of these factors were significantly associated with newly diagnosed infections in our cohort (all p-values > 0.05; see revised Table 2). For example, hypertension (OR 1.26, p = 0.54), diabetes (OR 1.13, p = 0.76), and CAD (OR 1.31, p = 0.49) were not predictive of infection. Although malignancy showed a borderline association (OR 7.27, p = 0.0516), it was present in only a small subset of patients and thus had limited influence on the overall model. Based on these results, we excluded these variables from our multivariable models, following the conventional approach of including only those covariates with p < 0.05 in univariable analyses to reduce model complexity and avoid overfitting—particularly important given the limited number of infection events. Instead, we prioritized simplified models that incorporated qSOFA and SIRS, which are bedside clinical scores that reflect the severity of systemic illness. These tools are widely used in emergency settings and offer practical utility for early risk stratification, especially when detailed comorbidity data may not be readily available.

• Biological Rationale for MDW Elevation: We have expanded the discussion to include a more detailed biological explanation for why MDW rises during infection. MDW quantifies the heterogeneity in monocyte size, which increases as monocytes become activated in response to infectious stimuli. During early stages of infection, circulating monocytes undergo nuclear deformation, cytoplasmic vacuolization, and membrane restructuring—processes driven by pro-inflammatory cytokines such as IL-6 and TNF-α. These changes lead to an increase in volumetric dispersion, which is detected by laser-based hematology analyzers as elevated MDW. Notably, MDW elevations can occur before overt leukocytosis or CRP elevation, positioning MDW as an early biomarker of innate immune activation. Furthermore, MDW complements systemic inflammatory markers like CRP and WBC count by capturing distinct cell-based dynamics of the host immune response.

• Recommendations for Future Research: In the revised Discussion (p. 23), we now emphasize the need for further investigation to validate the clinical utility of MDW. Specifically, prospective, multicenter studies are warranted to confirm the generalizability of MDW thresholds and models across diverse AMI populations. Moreover, randomized controlled trials assessing the impact of MDW-guided clinical interventions—such as earlier antibiotic initiation or escalation of care—could provide critical evidence regarding its potential to improve outcomes, including reduced length of stay and healthcare costs. Such studies would be essential before widespread clinical implementation of MDW-based infection surveillance in AMI patients undergoing PCI. (Please see pages 21–23 in the revised manuscript for detailed additions.)

6. Conclusions

Consider emphasizing how early detection using MDW-based models might improve patient outcomes (e.g., reduced LOS, timely antibiotic administration) and potentially lower healthcare costs.

Stress the need for further validation in larger, prospective, and multicenter studies before widespread clinical implementation.

• Thank you for this valuable comment. In the revised Conclusion section, we have emphasized the potential clinical and benefits of early infection detection using MDW-based models—such as enabling earlier antibiotic administration, reducing diagnostic uncertainty, minimizing delays in clinical decision-making, and potentially shortening hospital length of stay. We also stress the importance of further validation through larger, prospective, and multicenter studies to confirm the robustness and generalizability of our findings before these models can be implemented widely in clinical practice. (Please see page 23 in the revised manuscript.)

Responses to reviewer #2:

The manuscript of Sheng-Feng et al. reports a retros

---

## [Decision Letter · Decision Letter 1]

5 May 2025

PONE-D-25-04404R1Monocyte Distribution Width Enhances the Detection of Infection in Patients After Primary Percutaneous Coronary InterventionPLOS ONE

Dear Dr. Hou,

Thank you for submitting your manuscript to PLOS ONE. After careful consideration, we feel that it has merit but does not fully meet PLOS ONE’s publication criteria as it currently stands. Therefore, we invite you to submit a revised version of the manuscript that addresses the points raised during the review process.

**ACADEMIC EDITOR: ** In accord with Reviewer's suggestions, minor issues sould be edited point-by-point**.**

We look forward to receiving your revised manuscript.

Kind regards,

Vincenzo Lionetti, M.D., PhD

Academic Editor

PLOS ONE

Journal Requirements:

Reviewers' comments:

Reviewer's Responses to Questions

**Comments to the Author**

1. If the authors have adequately addressed your comments raised in a previous round of review and you feel that this manuscript is now acceptable for publication, you may indicate that here to bypass the “Comments to the Author” section, enter your conflict of interest statement in the “Confidential to Editor” section, and submit your "Accept" recommendation.

Reviewer #1: All comments have been addressed

Reviewer #2: All comments have been addressed

2. Is the manuscript technically sound, and do the data support the conclusions?

Reviewer #1: Yes

Reviewer #2: Yes

3. Has the statistical analysis been performed appropriately and rigorously? 

Reviewer #1: Yes

Reviewer #2: Yes

4. Have the authors made all data underlying the findings in their manuscript fully available?

Reviewer #1: Yes

Reviewer #2: Yes

5. Is the manuscript presented in an intelligible fashion and written in standard English?

Reviewer #1: Yes

Reviewer #2: Yes

6. Review Comments to the Author

Reviewer #1: (No Response)

Reviewer #2: a) On Tables 1 and 2:

1) Please check the SI units for the complete blood count (CBC) as follows:

- white blood cells (WBC), neutrophil, lymphocyte, and monocyte counts: ×10^9/L

- red blood cells (RBC): ×10^12/L

- hemoglobin (HGB): g/L

- hematocrit (HCT): L/L

2) Please delete mean CRP and NLR results because values were not-normally distributed (only median values need to be reported!). Moreover, check troponin T values distribution in the population studied (if values are not-normally distributed report median and IQ range in the text and in the Tables).

b) Methods

Please delete the words “Using a Hematology and Clinical

Chemistry Analyzer” from “Biomarkers measured using a hematology and clinical chemistry analyzer” paragraph because redundant.

c) Results

Edit the text on page 9 (“participant characteristics” paragraph) as follows:

…among all patients, the mean white blood cell (WBC) count was 18.1 ± 2.4 × 10^9/L (reference range: 4.0–10.0 ×10^9/L), the mean absolute neutrophil count was 7.48 ± 3.56 ×10^9/L (reference: 1.8–7.7 ×10^9/L), the mean lymphocyte count was 2.44 ± 1.62 ×10^9/L (reference: 1.0–3.2 ×10^9/L), the mean monocyte count was 0.77± 0.32 ×10^9/L (reference: 0.2–0.8×10^9/L),

and the MDW was 18.1 ± 2.4 unit (reference: <20 units). Regarding the

clinical chemistry tests, PCR reference value was < 5 mg/L, the

mean creatine kinase was 445.1 ± 878.1 unit/L (reference: 40–200 U/L), creatine kinase MB subunit was 47.3 ± 65.4 unit/L (reference: <25 U/L), and troponin-T reference value: <14 ng/L)…

7. PLOS authors have the option to publish the peer review history of their article (what does this mean? ). If published, this will include your full peer review and any attached files.

**Do you want your identity to be public for this peer review?** For information about this choice, including consent withdrawal, please see our Privacy Policy .

Reviewer #1: No

Reviewer #2: No

---

## [Author Response · Author response to Decision Letter 2]

6 May 2025

We sincerely thank the reviewers for their constructive and insightful comments. In response, we have revised the manuscript accordingly to address each of the points raised. All changes made to the original manuscript have been highlighted in red text. Our detailed responses to each comment are provided below:

Responses to reviewer #1:

(No Response)

• We sincerely thank Reviewer #1 for taking the time to review our manuscript.

Responses to reviewer #2:

a) On Tables 1 and 2:

1) Please check the SI units for the complete blood count (CBC) as follows:

- white blood cells (WBC), neutrophil, lymphocyte, and monocyte counts: ×10^9/L

- red blood cells (RBC): ×10^12/L

- hemoglobin (HGB): g/L

- hematocrit (HCT): L/L

• We thank the reviewer for this important suggestion. We have carefully revised the units for the complete blood count (CBC) in Tables 1 and 2 as requested. Specifically, WBC, neutrophil, lymphocyte, and monocyte counts are now reported as ×10⁹/L. Corresponding adjustments have also been made in the main text to ensure consistency. (Please see the revised Table 1, Table 2, and the text on page 9.)

2) Please delete mean CRP and NLR results because values were not-normally distributed (only median values need to be reported!). Moreover, check troponin T values distribution in the population studied (if values are not-normally distributed report median and IQ range in the text and in the Tables).

• We appreciate the reviewer’s attention to data presentation. Following the suggestion, we have deleted the mean and standard deviation (SD) values for CRP and NLR from both the text and Tables 1 and 2.

• CRP and NLR are now reported as median and interquartile range (IQR) only.

• We reassessed the distribution of troponin-T values using the Shapiro–Wilk test, which confirmed non-normality. Accordingly, troponin-T values are now reported as median and IQR in both the main text and tables. (Please see revised Table 1, Table 2)

b) Methods

Please delete the words “Using a Hematology and Clinical

Chemistry Analyzer” from “Biomarkers measured using a hematology and clinical chemistry analyzer” paragraph because redundant.

• Thank you for pointing this out. We have deleted the redundant phrase “using a hematology and clinical chemistry analyzer” from the Methods section to enhance clarity.

1. Please add to the references: Martinez-Iribarren A, Tejedor X, Sala Sanjaume A, Leis A, Dolade Botias M, Morales-Indiano C. Performance evaluation of the new hematology analyzer UniCel DxH 900. Int J Lab Hematol 2021;43:623-31 where DxH 900 analyzer showed good analytical performance in measuring MDW, the novel parameter for the early diagnosis of sepsis studied by Sheng-Feng et al.

c) Results

Edit the text on page 9 (“participant characteristics” paragraph) as follows:

…among all patients, the mean white blood cell (WBC) count was 18.1 ± 2.4 × 10^9/L (reference range: 4.0–10.0 ×10^9/L), the mean absolute neutrophil count was 7.48 ± 3.56 ×10^9/L (reference: 1.8–7.7 ×10^9/L), the mean lymphocyte count was 2.44 ± 1.62 ×10^9/L (reference: 1.0–3.2 ×10^9/L), the mean monocyte count was 0.77± 0.32 ×10^9/L (reference: 0.2–0.8×10^9/L),

and the MDW was 18.1 ± 2.4 unit (reference: <20 units). Regarding the

clinical chemistry tests, PCR reference value was < 5 mg/L, the

mean creatine kinase was 445.1 ± 878.1 unit/L (reference: 40–200 U/L), creatine kinase MB subunit was 47.3 ± 65.4 unit/L (reference: <25 U/L), and troponin-T reference value: <14 ng/L)…

• We thank the reviewer for providing the specific suggested wording. We have edited the “Participant Characteristics” paragraph in the Results section according to the reviewer’s instructions to ensure proper unit usage, consistent format, and improved clarity.

---

## [Editor Report · Decision Letter 2]

12 May 2025

Monocyte Distribution Width Enhances the Detection of Infection in Patients After Primary Percutaneous Coronary Intervention

PONE-D-25-04404R2

Dear Dr. Hou,

We’re pleased to inform you that your manuscript has been judged scientifically suitable for publication and will be formally accepted for publication once it meets all outstanding technical requirements.

Kind regards,

Vincenzo Lionetti, M.D., PhD

Academic Editor

PLOS ONE
---

## [Editor Report · Acceptance letter]

PONE-D-25-04404R2

PLOS ONE

Dear Dr. Hou,

I'm pleased to inform you that your manuscript has been deemed suitable for publication in PLOS ONE. Congratulations! Your manuscript is now being handed over to our production team.

Kind regards,

on behalf of

Prof. Vincenzo Lionetti

Academic Editor

PLOS ONE